# Measuring the Image of Private University as a Generic Product: Validation of a Scale

**Purificación Alcaide-Pulido** [1,*] , **Belén Gutiérrez-Villar** [2] **and Mariano Carbonero-Ruz** [3]

1    Communication and Education Department, Universidad Loyola Andalucía,
     Escritor Castilla Aguayo, 4, 14004 Córdoba, Spain
2    Management Department, Universidad Loyola Andalucía, Escritor Castilla Aguayo, 4, 14004 Córdoba, Spain
3    Quantitative Methods Department, Universidad Loyola Andalucía, Escritor Castilla Aguayo, 4,
     14004 Córdoba, Spain
*    Correspondence: palcaide@uloyola.es

**Abstract:** The compulsory nature of online training in university education, brought about by COVID-19, has opened the door to the emergence of several potential competitors in the university space. Thus, measuring a university's image may have even greater importance for the management and differentiation of universities in the new post-COVID-19 horizon. This study aims to test whether a standardized scale of brand image measurement is valid for measuring the image of the "private university" product. A non-probabilistic convenience sample was chosen, collecting information from 728 citizens from the same territory (Andalusia). The procedure to validate the scale involves dividing the sample (728) into two sub-sets: one to establish the scale, and the other to validate the results. The methodology applied is Confirmatory Factor Analysis using EQS 6.3 software. The scale was validated, and the main results show that people favor the quality of private universities, their commitment to society, and the perfect option that they are. Additionally, results show the idea that private universities present characteristics absent from public ones as non-significant, and do not agree that private universities provide a high value concerning the price that has to be paid.

**Keywords:** corporate image; higher education; CFA; branding management; post-COVID-19 pandemic

## 1. Introduction

In the competitive and changing context of Higher Education Institutions (HEIs) after the COVID-19 pandemic, coupled with internationalization and the ease of mobility, inevitably, there has been an increase in competition manifested in the emergence of the incorporation of distance and open education models (Cunha et al. 2020; Nuseir et al. 2021), which causes a greater need for better business management (Adi et al. 2021). One of the critical resources for more efficient use of these institutions' capacities is their image (Chapleo 2011; de Heer and Tandoh-Offin 2015; Schlesinger et al. 2017; Lafuente-Ruiz-de-Sabando et al. 2018; Sahin 2021; Alcaide-Pulido et al. 2021); leading to the importance of being able to measure it appropriately (Alcaide-Pulido et al. 2017; Salimi and Abdi 2018) and the readiness to change (Alolabi et al. 2021).

Image is an essential subject in academic research concerning marketing. There is agreement that image can be measured, but it takes work to measure Kazoleas et al. (2001). Several authors support the idea that the image of a product, brand, or company are subjective phenomena that can be measured by collecting the opinions of individuals or groups without the need for them to be consumers (Schlesinger et al. 2017; Keller 1993). However, there is no generally accepted model for measuring perceptual images or the methodologies proposed, and the variables used in published models vary (Stern et al. 2001).

In line with the above, this study aims to contribute further evidence to the discussion on the perceptual measurement of university image with one objective: it was decided

to examine a standardized scale for measuring brand image proposed by Salinas et al. (2004), consisting of three dimensions, functional image, affective image, and reputation, to confirm its validity when the object measured is the private university, without specifying a specific brand.

Nevertheless, this is not the only contribution of the research. Carrying out the study on the population in general, following some of the main studies that have begun to analyze university image in such a context (Landrum et al. 1999; Kazoleas et al. 2001; Arpan et al. 2003; Sahin 2021), thus, not only members of the university community, as is usual in this type of study (Wilkins and Huisman 2015; Gutiérrez-Villar et al. 2017), confirmed the representativeness and relevance of this research.

A final contribution we consider exciting lies in the methodology used, since our objective involves the validation of a scale mainly through the unusual procedure of dividing the sample into two sub-sets: one for training (forming the scale) and the other for testing (validation of the results). This practice is common in some areas of research (Ólafsson et al. 2011; Alderman and Headey 2017; Bjelica et al. 2019), but is less so in the area of marketing management (Xue and Hong 2016). In addition, the confirmation of the homogeneity of both subsamples proposed by Lung-Yut-Fong et al. (2015) is relatively new.

In most countries (as is the case in Spain), public and private universities coexist; focusing the study on the issue of measuring the image of private universities is considered pertinent, since this issue is debated in society and it is easy for all individuals to give their opinion on the subject.

Specifically, the Spanish University System was made up of a total of 83 universities in the 2019–2020 academic year; 50 public and 33 private. However, private universities in Spain are evenly distributed throughout the country; for example, there is a more significant number of them around Madrid, the country's capital, and there are only four autonomous communities that do not have any in their territory. In the case of Andalusia, there is only one private university (Spanish University System 2022).

The objective of this study is to validate an image measurement scale for the case of private universities; no attempt has been made to appreciate differences based on geographic or other sociodemographic characteristics. Given this non-homogeneous distribution of private universities and given the budgetary limitations that prevented us from covering the entire nation, it was decided to focus the collection of information on citizens of the same autonomous territory, which turned out to be Andalusia due to the proximity of the researchers to this area.

The remainder of the work is structured in four sections. First, we develop the conceptual framework on the importance of measuring the image for HEIs and the models for measuring brand image. Next, we describe the questionnaire structure with suitable modifications for the study context. Then we explain the methods used and show the results of the data analysis. Finally, we present the conclusions, implications, limitations, and future lines of research for the management of educational institutions, especially in the private sector.

## 2. Literature Review

In marketing, it is generally accepted that in a competitive marke it is necessary to know and measure the product's image, precisely, brand image (de Heer and Tandoh-Offin 2015; Schlesinger et al. 2017; Meirinhos et al. 2022). This need is even more significant in the for-profit sector, where interest in measuring brands' image becomes a strategic objective, which has extended over recent years to sectors such as higher education (Chapleo 2011; Pavlova et al. 2019).

HEIs recognize the importance of their image as a tool for positioning in a competitive market (Chapleo 2011; Baltaru 2019; Pavlova et al. 2019; Alhaza et al. 2021; Sahin 2021). Specifically, since establishing the Bologna Process in the European Higher Education Area (Bologna 1999), HEIs have gained importance and have been positioned in the European community as centers of R + D + i and learning, created and maintained by society aiming

to achieve prosperity for the population through generating knowledge (de Heer and Tandoh-Offin 2015; Pavlova et al. 2019; Esposito et al. 2021).

The emergence of new universities, both public and private, in most countries and with the corresponding impact on social and economic systems (Osman et al. 2018; Orazbayeva et al. 2019), or even the incorporation of distance and open education models in existing universities after the Covid-19 pandemic (Cunha et al. 2020; Nuseir et al. 2021), has meant these institutions seek to stand out from the competition (de Heer and Tandoh-Offin 2015; Schlesinger et al. 2017; Alcaide-Pulido et al. 2017; Sahin 2021). Increasingly, public and private HEIs operate in the same economic system and must improve their communication strategies to remain in the system (Alhaza et al. 2021; Sahin 2021), which becomes a challenge. All of this competition, the connection to society, the need for differentiation and positioning, and the challenges are associated with image, so its measurement has become a fundamental aspect of the management of these institutions (Chapleo 2011; Alcaide-Pulido et al. 2017; Lafuente-Ruiz-de-Sabando et al. 2018; Pavlova et al. 2019; Alhaza et al. 2021).

Functional aspects appear in the studies, and we find work using affective components among psychological ones. So, to measure an image, it is necessary to resort to multifactor models that combine tangible attributes or benefits with others of a psychological nature. In contrast, others find it necessary to add reputation (Lafuente-Ruiz-de-Sabando et al. 2018; Rodríguez-Díaz et al. 2019; Gutiérrez-Villar et al. 2021; Meirinhos et al. 2022). In this analysis, the authors confirm that the interaction between image and reputation helps explain a critical aspect in the marketing of this institution: loyalty (Lafuente-Ruiz-de-Sabando et al. 2018; Del-Castillo-Feito et al. 2019).

To be able to unite this multiplicity of factors, for all the proposals reviewed in this research, it was decided to resort to the model to measure image by Salinas et al. (2004), including some improvements to the scale one year later (Martínez et al. 2005). The questionnaire is structured around three factors related to tangible attributes or benefits (functional or cognitive image), more emotive aspects (affective image), and the general symbolic social perception (reputation). In the original proposal, a scale of nine items was used, three for each construct.

After reviewing the literature of different studies on university image, another characteristic extracted is that most of the collectives investigated belong to the university environment. They present a predominance of studies focusing on students (Arpan et al. 2003; Veloutsou et al. 2004; Osman et al. 2018; Salimi and Abdi 2018; Alhaza et al. 2021; Sahin 2021); and, to a lesser extent, university staff (Luque and del Barrio 2008), pre-university students (Gutiérrez-Villar et al. 2017), PhD students (Pavlova et al. 2019), or graduates (Palacio et al. 2002). None were found to study the university's image in Spanish society, and only three of the first studies on the university image used a sample of the population in general (Landrum et al. 1999; Kazoleas et al. 2001; Arpan et al. 2003). So, the theoretical review confirms the relevance of the study presented here, which aims to validate a scale to measure the image on citizenship regardless of whether or not he or she has studied at the university level.

## 3. Materials and Methods

### 3.1. Sample

The type of sampling used was intentional. The questionnaire launching method was online through a consolidated consumer panel, managed by a marketing research company with consistent control of registered persons. The stability and maturity implied by its past will have transferred to the sample, strengthening its representativeness. The initial sample consisted of 778 questionnaires. With our limited budget, this number was the maximum we could afford. The initial sample was filtered, eliminating atypical observations, which resulted in 50 questionnaires being eliminated.

In total, 728 Andalusians validly completed the questionnaire. The sample was 446 men (61.4%) and 281 women (38.6%) between 18 and 65 years old. The distribution by age bracket was equal in the four age brackets chosen: 182 persons aged 18 to 25 (25%),

182 persons aged 26 to 35 (25%), 182 persons aged 46 to 55 (25%) and 182 persons aged 56 to 65 (25%).

*3.2. Instrument*

As already indicated in the introduction, this study aims to validate a standardized image measurement scale for the case of private universities instead of generating a scale created expressly for the university sector. The scale chosen was originally proposed by Salinas et al. (2004).

This standardized scale has been applied to different brands in various sectors of consumer goods (Martínez et al. 2005, 2007; Buil et al. 2008). In all of these studies, the scale was validated, resulting in explaining the three constructs (functional, affective, and reputation), although, as the authors themselves state, the nine items originally proposed to assess brands in the car sector should be adapted if applied to brands in other sectors, and can be extended or substituted with others, as long as respecting the three-dimensional structure.

Based on the results of the studies mentioned above, the items of the model were adapted to the specific issue studied here, resulting in an initial questionnaire formed of 14 questions, five associated with each of the first two factors, and four with the third, as shown in Table 1.

**Table 1.** Scale to measure the private university's image.

| Functional Factor | Affective Factor | Reputational Factor |
|---|---|---|
| F1. The education and training in the private university are of high quality. | A1. The private university arouses affection. | R1. The private university is the best option. |
| F2. The private university's facilities are of high quality. | A2. The private university conveys a personality that differentiates it from public ones. | R2. The private university is committed to society. |
| F3. The research coming from private universities is of high quality. | A3. Enrolling in a private university says something about your type of person. | R3. The private university is very consolidated in the market. |
| F4. Private universities have characteristics that public ones do not have. | A4. The private university does not disappoint its customers. | R4. Private universities are placed very highly in university rankings. |
| F5. The private university provides high value regarding the price that must be paid. | A5. The private university has a differentiated ideological component. | |

Source: own elaboration adapted from Martínez et al. (2005).

As for formulation of the questions, each one is a statement and respondents are asked to indicate their level of agreement or disagreement on a 7-point Likert scale (1 = complete disagreement, 7 = complete agreement). The questionnaire ends with the respondent's socio-demographic characteristics, and some questions related to their level of education and assessment.

The initial model (Model 1) for this first stage is shown in Figure 1, accurately reflecting the structure of the questionnaire, organized as indicated when describing it, in three factors (Functional, Reputational, and Affective), and adding the possibility of there being correlations between them.

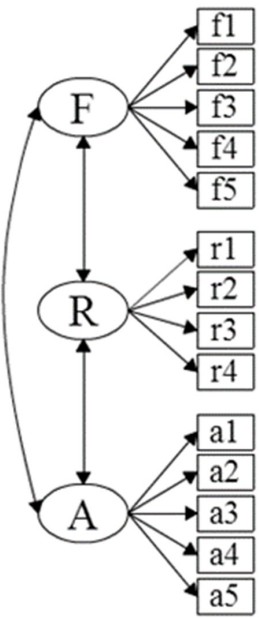

**Figure 1.** Model 1 or initial model.

### 3.3. Data Collection and Analysis Procedure

Factor Analysis is a technique of grouping a set of variables into another smaller one, whose elements are generally classed as factors. This reduction in size is usually performed in one of two diametrically opposite situations: at one extreme, complete uncertainty regarding the number of factors expected, and even more so regarding each one's composition. It is the field of applying Exploratory Factor Analysis (EFA).

Regarding insignificant variables, we eliminated from the research two variables that did not emerge as significant in Exploratory Factor Analysis (EFA). Variable a3 is related to the type of person who attends a private university, and variable a5 is related to the ideological character of the university.

At the opposite extremes are cases where the underlying factor structure is assumed, that is, how the variables will be grouped in the process. In these cases, the analysis serves to confirm (or reject) that structure; this task is the function of Confirmatory Factor Analysis (CFA) (Hoyle 2000, p. 465). In our case, neither of the extremes matches the problem studied, which is halfway between them: previous research indicates a partially known but not unequivocally established structure, which means it is not appropriate to adopt either of the traditional focuses.

In these circumstances, as a recent method, we chose to focus on two steps based on CFA, which we believe comes closest to the issue as a starting point. In the first stage, a three-factor model was estimated: functional, affective, and reputational, arising from the studies by Salinas et al. (2004) and Martínez et al. (2005), if the structure given to the questionnaire is similar to the real one, but not necessarily identical.

For the first step of the analysis CFA, the 728 individuals were divided into two equal parts, each used in the two phases of constructing the model. This decision was made since using the same observations for construction and validation would not be advisable. This way, the second stage would not be independent of the first. That is, if to evaluate the model we use the same observations as in building it; we are likely to obtain a good result due to the parameters of the model having been estimated with the same sample as used for subsequent validation (Hurst et al. 2010; Zhen et al. 2013). Therefore, dividing a sample into subsamples of homogeneous composition is crucial to ensure that the analyses they are used in have general validity, and not only for the type of subsample used.

Among the few methods available to divide the sample, as this is a very recent technique, the method developed by Lung-Yut-Fong et al. (2015) was chosen. These authors performed an extension to the multivariate case of the Wilcoxon-Mann-Whitney univariate

contrast of equality of means without normality requirements, the so-called *t*-test. The difference is applied to successive sample partitions in two randomly obtained subsets in dividing the sample. Similarity is assessed through a statistic of known approximate distribution (specifically chi-squared) until the sub-samples obtained are statistically homogeneous for a chosen level of significance, 5% in this case (Lung-Yut-Fong et al. 2015).

*3.4. Quantitative Analysis*

Applying the procedure to the initial sample led to two sub-samples of 363 and 365, whose associated *p*-value is 0.948, very close to one. We accept the division, as the similarity is assured to a great extent. The first subsample will be used to confirm the model, and the second to validate it.

After estimating Model 1 with the first sub-sample formed of 363 individuals (Figure 1), this was re-estimated using Lagrange multipliers to obtain a satisfactory definitive model from the point of view of adjustment.

In the following stage, the estimated and modified model was subject to assessment using the second subsample formed of 365 individuals. It determined whether the goodness-of-fit achieved in the previous stage was inherent to the model or adapted exclusively to the sample it had obtained (overtraining). In both cases, the results were satisfactory.

To estimate and validate both models of Confirmatory Factor Analysis, we used EQS 6.3 software, choosing maximum likelihood (ML) as the method to estimate the parameters. ML is a widely used method to evaluate this model type when the intention is to find parameter estimations that allow a more effective occurrence of the sample obtained, maximizing its likelihood (Hoyle 2000). It is also a very robust method of analysis that can be applied when there is no condition of univariate or multivariate normality of data (as in this case); without severe losses in the theoretical properties of the estimations obtained (Hair et al. 2006).

## 4. Results

*4.1. Correlation Analysis*

Examination of the data is a necessary step, and when careful analysis of the data is performed, better prediction and more accurate assessment of dimensionality is achieved. Outliers, or extreme responses, can unduly influence the outcome of a multivariate analysis (Hair et al. 2006), so before starting the factor analysis it is necessary to detect outliers in the base by calculating the Mahalanobis distance and eliminating those with a *p*-value less than one thousandth. This leaves 728 records.

Once the outliers have been detected, the degree of correlation between the variables is analyzed, where the correlation matrix is adequate, only p6_3 and p6_5 show very low correlations with p12_3, and in general quite low correlations with the rest of the variables, so they are eliminated (Table 2).

The variable PARTIES is generated, which takes the value 1 if in Q12 4 or more are answered, and 0 if 3 or fewer are answered. The division placed both groups in almost the same proportion: 49.1% of non-partisans versus 50.9% of supporters. In relation to socio-demographic variables, with respect to sex there is a relationship: men are more likely to be supporters than women; and also with respect to age: the older they are, the more likely they are to be supporters (Appendix A).

**Table 2.** Correlation analysis between variables.

|       | P12_3 | Pf_1 | Pf_2 | Pf_3 | Pf_4 | Pf_5 | Pa_1 | Pa_2 | Pa_3 | Pa_4 | Pa_5 | Pr_1 | Pr_2 | Pr_3 | Pr_4 |
|-------|-------|------|------|------|------|------|------|------|------|------|------|------|------|------|------|
| P12_3 | 1 | 0.549 | 0.386 | 0.495 | 0.489 | 0.581 | 0.615 | 0.535 | 0.269 | 0.571 | 0.221 | 0.733 | 0.705 | 0.588 | 0.675 |
| Pf_1 | 0.549 | 1 | 0.688 | 0.691 | 0.556 | 0.589 | 0.557 | 0.522 | 0.236 | 0.489 | 0.282 | 0.519 | 0.500 | 0.482 | 0.593 |
| Pf_2 | 0.386 | 0.688 | 1 | 0.613 | 0.569 | 0.482 | 0.407 | 0.486 | 0.287 | 0.422 | 0.284 | 0.328 | 0.377 | 0.401 | 0.480 |
| Pf_3 | 0.495 | 0.691 | 0.613 | 1 | 0.521 | 0.597 | 0.529 | 0.476 | 0.267 | 0.456 | 0.257 | 0.486 | 0.509 | 0.478 | 0.594 |
| Pf_4 | 0.489 | 0.556 | 0.569 | 0.521 | 1 | 0.611 | 0.468 | 0.601 | 0.276 | 0.436 | 0.269 | 0.507 | 0.481 | 0.430 | 0.531 |
| Pf_5 | 0.581 | 0.589 | 0.482 | 0.597 | 0.611 | 1 | 0.599 | 0.547 | 0.284 | 0.541 | 0.196 | 0.577 | 0.573 | 0.471 | 0.608 |
| Pa_1 | 0.615 | 0.557 | 0.407 | 0.529 | 0.468 | 0.599 | 1 | 0.594 | 0.271 | 0.557 | 0.216 | 0.611 | 0.661 | 0.571 | 0.608 |
| Pa_2 | 0.535 | 0.522 | 0.486 | 0.476 | 0.601 | 0.547 | 0.594 | 1 | 0.436 | 0.545 | 0.360 | 0.562 | 0.572 | 0.541 | 0.622 |
| Pa_3 | 0.269 | 0.236 | 0.287 | 0.267 | 0.276 | 0.284 | 0.271 | 0.436 | 1 | 0.483 | 0.424 | 0.355 | 0.337 | 0.374 | 0.389 |
| Pa_4 | 0.571 | 0.489 | 0.422 | 0.456 | 0.436 | 0.541 | 0.557 | 0.545 | 0.483 | 1 | 0.338 | 0.609 | 0.587 | 0.551 | 0.588 |
| Pa_5 | 0.221 | 0.282 | 0.284 | 0.257 | 0.269 | 0.196 | 0.216 | 0.360 | 0.424 | 0.338 | 1 | 0.249 | 0.199 | 0.315 | 0.315 |
| Pr_1 | 0.733 | 0.519 | 0.328 | 0.486 | 0.507 | 0.577 | 0.611 | 0.562 | 0.355 | 0.609 | 0.249 | 1 | 0.762 | 0.631 | 0.700 |
| Pr_2 | 0.705 | 0.500 | 0.377 | 0.509 | 0.481 | 0.573 | 0.661 | 0.572 | 0.337 | 0.587 | 0.199 | 0.762 | 1 | 0.720 | 0.698 |
| Pr_3 | 0.588 | 0.482 | 0.401 | 0.478 | 0.430 | 0.471 | 0.571 | 0.541 | 0.374 | 0.551 | 0.315 | 0.631 | 0.720 | 1 | 0.733 |
| Pr_4 | 0.675 | 0.593 | 0.480 | 0.594 | 0.531 | 0.608 | 0.608 | 0.622 | 0.389 | 0.588 | 0.315 | 0.700 | 0.698 | 0.733 | 1 |

Source: SPSS.

Finally, the assumptions underlying most multivariate analyses are reviewed. The most important aspect of the results is the evaluation of the model's goodness of fit. Here, we obtained some of the usual indices in this type of analysis, namely the comparative fit index (CFI)[1], the normed fit index (NFI), the non-normed fit index (NNFI)[2] and the root mean square error of approximation (RMSEA)[3] (Hu and Bentler 1999).

### 4.2. First Stage: Validation of the Model with the First Sub-Sample

The values of the indicators in the initial estimation (CFI = 0.919, NNFI = 0.895, NFI = 0.905 and RMSEA = 0.119) lead to rejecting its validity. However, three of the four indicators show values close to validity, and the flexibility of the model's theoretical grounding led us to analyze its improvements, deduced from the modification indices and the Lagrange test. Reviewing the Lagrange multivariate multipliers indicates that significant improvement can be obtained if admitting the possibility that some of the indicators could be linked to more than one factor. After reviewing the significance of the potentially more efficient changes as regards improved goodness-of-fit, we decided to incorporate the two that seem most justified theoretically: if variable f5: "The private university provides high value about the price that has to be paid", also has an affective component, that variable r3: "The private university is very consolidated in the market" is also related to functional elements.

Carrying out these changes, and despite observing improvement in the fit indicators (CFI = 0.933, NNFI = 0.910, NFI = 0.920 and RMSEA = 0.111), we felt the model should be improved, above all due to the high value of RMSEA. A new analysis of the Lagrange multipliers shows the possible improvement linked to variable f4: "Private universities present characteristics that public ones do not have" with three factors: functional, to which it belongs, affective and reputational. We consider this possibility equivalent to stating that by loading on all factors, the variable is insignificant. Therefore, we decided to eliminate it from the model.

Eliminating the indicated variables and the modifications led to obtaining an improved model that we can consider definitive depending on its validation, as shown in Figure 2.

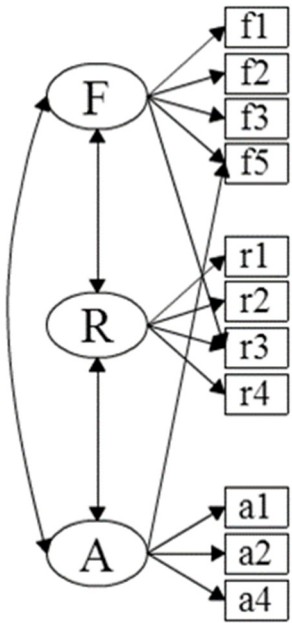

**Figure 2.** Model 2 or improved model.

The indicators of the second model (CFI = 0.965, NNFI = 0.950, NFI = 0.952 and RMSEA = 0.086) are very satisfactory. So, we consider this the definitive model for its validation with the rest of the sample.

Figure 2 presents how the variables are related and how they load on the factors once the changes estimated with the CFA results are made. We can see that there are two factors (f5 and a3) that load on two first-order factors: f5 (The private university provides high value regarding the price that must be paid), which naturally belongs to the functional image factor, and loads on the affective factor. The variable r3 (The private university is very consolidated in the market), which by its nature belongs to the reputational factor, and the model's improvement would also load on the functional factor.

*4.3. Second Stage: Validation of the Results with the Second Sub-Sample*

The expectations for the second stage, supposing validation of the model, are stability, in that the model proposed, without new modifications, should be acceptable. Furthermore, if the structure detected in the first stage is correct, both the global adjustment measures and the factor loading should be similar in both estimations.

Tables 3 and 4, respectively, present the goodness-of-fit estimators and the values estimated for the parameters of Model 2 with the results of the first sub-sample (363 individuals) and Model 2 checked with the second sub-sample (365 individuals).

**Table 3.** Global measurement.

|  | Model 1. First Sub-Sample | Model 2. Second Sub-Sample |
|---|---|---|
| CFI | 0.965 | 0.963 |
| NNFI | 0.950 | 0.947 |
| NFI | 0.952 | 0.951 |
| RMSEA | 0.086 | 0.088 |

Source: own elaboration from results.

The comparison between the indicators of the two models suggests that, although the structure has been estimated from a set of values and partially adapted to them, the adequate degree of reliability obtained is not the result of this adaptation. The indicators turn out to be practically identical for both samples. To validate the model estimated, it is not enough for the reliability indicators to coincide since the loadings estimated should also

do so. Table 4 presents the comparison of the models, including the correlations estimated. Once again, the coincidence is found to be generally very high.

**Table 4.** Estimated parameters.

| Factor | Variable | Model 1 | Model 2 |
|--------|----------|---------|---------|
| F | F1 | 0.865 ** | 0.883 ** |
|   | F2 | 0.776 ** | 0.777 ** |
|   | F3 | 0.816 ** | 0.838 ** |
|   | F5 | 0.286 ** | 0.479 ** |
|   | R3 | 0.176 ** | 0.358 ** |
| R | R1 | 0.852 ** | 0.854 ** |
|   | R2 | 0.892 ** | 0.903 ** |
|   | R3 | 0.835 ** | 0.827 ** |
|   | R4 | 0.754 ** | 0.537 ** |
|   | F5 | 0.53 ** | 0.341 ** |
| A | A1 | 0.82 ** | 0.785 ** |
|   | A2 | 0.755 ** | 0.769 ** |
|   | A4 | 0.772 ** | 0.732 ** |
|   | Cov(F,R) | 0.631 | 0.708 |
|   | Cov(F,A) | 0.766 | 0.822 |
|   | Cov(R,A) | 0.901 | 0.926 |

Source: own elaboration from results. ** $p < 0.001$.

## 5. Discussion

The standardized three-factor scale has been validated. However, some variables are not significant for the analysis (a5 and a3), and others are significant in dimensions other than the one originally proposed.

The improvement of the model with the first confirmatory analysis models detects that three variables load on more than one factor. These are f5, r3, and f4. The variable f5 is "the private university provides a high value to the detriment of the price to be paid for it", and the other variable r3 is "the private university is very consolidated in the market". Results indicate that both variables load on the functional and affective dimensions.

In addition, the variable f4 indicates that private universities present characteristics that public ones do not have. Results show it loads on the three factors: functional, to which it belongs, affective, and reputational. Also, from a reputational point of view, they must emphasize the promotion of an image of commitment to society. We consider this possibility equivalent to stating that the variable is insignificant by loading on all factors. Therefore, we decided to eliminate it from the model because the statement that private universities have characteristics absent from public universities is not significant.

The improvement results of the second model by parameter comparison, including the correlations between factors (Table 3), show improvement in the second model in all variables except r4 and a1. The r4 is related to high positions of private universities in university rankings, while a1 determines that the private university causes affection. As there are only two variables, and they are also statistically significant, we can determine that model 2 is better than model 1.

The fact that it is the coefficients corresponding to the variables re-situated during the modification process ending the first stage that presents more significant divergence leaves, however, the uncertainty about the initial three-factor scheme. Another important finding is the high correlation between the so-called affective and reputational factors.

## 6. Conclusions and Limitations

Brand image is considered an asset through which companies can present differentiation from their rivals using appropriate measurement techniques that are operational

and can be applied to different contexts. This study proposes and validates the scale for measuring brand image submitted by Salinas et al. (2004), combined with the scale improved by these authors one year later (Martínez et al. 2005). These authors determined that the perceptions of a brand are grouped around three basic dimensions: functional, affective, and reputational.

As demonstrated with brands belonging to different sectors in earlier studies, the three factors, or dimensions, are statistically valid. Although the scale proposed can be applied to any marketing area, this study investigated the application to the perception of the private university in Andalusian society as a generic product in a general population. Thus, the general nature of the sample, regarding both the population (Andalusia) and the service analyzed (the private university), gives the study a distinctive characteristic. Concerning the participants, it is also essential to point out the methodology used, involving the sub-division of the sample into two parts used to estimate and subsequently validate the model.

In the first part of the analysis, EFA resolves the necessity to delete two variables. Variable a3 states that enrolling in a private university determines the type of person you are, and variable a5, in which it is analyzed that private universities have a differentiated ideological component. Several current studies show the need to promote the development of intercultural educational praxis as a skill in universities. It is necessary that governments, in general, and HEIs specifically integrate it into their policies to preserve cultural authenticity, human values, and cultural respect against racism and xenophobia. In particular, it is essential that teachers, as persons in charge of accompanying the education of the future, promote these practices of respect to generate understanding and justice inequality between groups of different cultures (Orozco-Vargas et al. 2020).

The results of this research may help decision-makers in private universities to find elements of differentiation in a university environment "revolutionized" by the increasingly competitive COVID-19 pandemic.

Thus, from both sides of the coin, the following have been identified as key functional aspects that private universities should promote: the image of quality in their teaching, in their facilities, and in their research to help society. Lastly, from an emotional point of view, they must enhance their image as an entity that does not disappoint those who choose it.

On the flip side of the coin, the question that proposed the idea of "private universities have characteristics that public ones do not have" has not turned out to be significant. This could be indicative that, in the case of Spain, citizens do not perceive major differences between private and public universities in functional attributes, such as facilities, research, or teaching, and the differences are more linked to reputational or affective aspects. These results may coincide with other researchers who state that broadly speaking, the strengths of the Spanish public university are the variety of degrees, the research programs developed, the international prestige of some of them, and lower public prices. On the other hand, the advantages associated with private universities are related to more personalized treatment of the student (Algaba Garrido 2015).

The affective attributes connected to brand personality are also considered, since many organizations' positioning strategies are supported by symbolic and emotive characteristics. As for the university's functional characteristics, the respondent dramatically appreciates the quality of education in private universities and considers the research emanating from private universities as being of excellent quality. These institutions, public and private, seek to differentiate themselves from the competition (de Heer and Tandoh-Offin 2015; Schlesinger et al. 2017; Alcaide-Pulido et al. 2017; Sahin 2021).

It will be interesting, in future studies, to contrast whether these results coincide in different countries.

In addition, following the results of this research, it also seems interesting to carry out future research to evaluate whether a simplified model, in which only two factors are used, one related to tangible aspects and the other related to affective aspects, could be useful for measuring the image of generic products, without an associated brand.



The main limitation of this study is the sample, which has been limited to a single territory within Spain. It would be helpful to replicate it in different geographic areas to see if the proximity to a more significant number of private universities could alter the results.

**Author Contributions:** Conceptualization, P.A.-P. and B.G.-V.; methodology, M.C.-R. and B.G.-V.; software, M.C.-R.; validation, P.A.-P., B.G.-V. and M.C.-R.; formal analysis, B.G.-V.; investigation, P.A.-P.; resources, P.A.-P.; data curation, M.C.-R.; writing—original draft preparation, P.A.-P.; writing—review and editing, B.G.-V.; visualization, P.A.-P. and B.G.-V.; supervision, M.C.-R.; project administration, P.A.-P. All authors have read and agreed to the published version of the manuscript.

**Funding:** This research received no external funding.

**Institutional Review Board Statement:** The study was conducted in accordance with the Declaration of Helsinki and approved by the Ethics Committee of Universidad Loyola Andalucía (September 2021) for studies involving humans.

**Informed Consent Statement:** Informed consent was obtained from all subjects involved in the study. Written informed consent has been obtained from the patient(s) to publish this paper.

**Data Availability Statement:** Not applicable.

**Conflicts of Interest:** The authors declare no conflict of interest.

## Appendix A

**Table A1.** Contingence table (Item: Gender).

| | | | Gender | | Total |
|---|---|---|---|---|---|
| | | | Men | Women | |
| Parties | No | Count | 199 | 170 | 369 |
| | | Expected frequency | 226.5 | 142.5 | 369.0 |
| | Sí | Count | 262 | 120 | 382 |
| | | Expected frequency | 234.5 | 147.5 | 382.0 |
| Total | | Count | 461 | 290 | 751 |
| | | Expected frequency | 461.0 | 290.0 | 751.0 |

**Table A2.** Chi-square test (Gender).

| | Value | gl | Asymptotic Sig. (Bilateral) |
|---|---|---|---|
| Pearson's Chi-square | 17.010 [a] | 1 | 0.000 |
| Continuity correction [b] | 16.398 | 1 | 0.000 |
| Likelihood ratio | 17.075 | 1 | 0.000 |
| Fisher's exact statistic | | | |
| Lineal by lineal association | 16.988 | 1 | 0.000 |
| N of valid cases | 751 | | |

[a] Indicates this is a statistical significance of the relationship. [b] The value of it indicates that is significant.

**Table A3.** Contingence table (Item: Age).

| | | | Age | | | | | Total |
|---|---|---|---|---|---|---|---|---|
| | | | 18–25 Years | 26–35 Years | 36–45 Years | 46–55 Years | 56–65 Years | |
| Parties | No | Count | 103 | 88 | 75 | 50 | 53 | 369 |
| | | Expected frequency | 73.7 | 73.7 | 73.7 | 73.7 | 74.2 | 369.0 |
| | Sí | Count | 47 | 62 | 75 | 100 | 98 | 382 |
| | | Expected frequency | 76.3 | 76.3 | 76.3 | 76.3 | 76.8 | 382.0 |
| Total | | Count | 150 | 150 | 150 | 150 | 151 | 751 |
| | | Expected frequency | 150.0 | 150.0 | 150.0 | 150.0 | 151.0 | 751.0 |

**Table A4.** Chi-square test (Age).

| | Value | gl | Asymptotic Sig. (Bilateral) |
|---|---|---|---|
| Pearson's Chi-square | 55.282 [a] | 4 | 0.000 |
| Continuity correction | 56.332 | 4 | 0.000 |
| Likelihood ratio | 51.320 | 1 | 0.000 |
| N of valid cases | 751 | | |

[a] Indicates this is a statistical significance of the relationship.

## Notes

[1]    CFI is a measure comparing the adjustment obtained for the model estimated with the assumption that there was no relation between the variables used. Its values range between 0 (no adjustment of the model to the data) and 1 (perfect adjustment of the model), taking 0.9 as the threshold of appropriate adjustment.

[2]    The NFI and NNFI indices are similar measures that situate the estimated model on a scale with the extremes of null model and perfect fit, differing in considering the models' degrees of freedom or not. Both because CFI has values between 0 and 1, considering those with the highest values in these indices as best-adjusted models.

[3]    Finally, RMSEA, as a measure of the errors made, should be interpreted in the opposite direction, with the most appropriate models being those in which this indicator has a lower value, a standard reference being a figure not above 0.06.

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
