# Peer review of "Measuring the Image of Private University as a Generic Product: Validation of a Scale"

_admsci, doi:10.3390/admsci12040178_

Round 1

Reviewer 1 Report

The manuscript presented raises a question that could be interesting if the analysis had been broader, not limited to a Spanish autonomous community in which, at least so far, there is only one private university.  For this reason, I consider it of little relevance.  

On the other hand, the text focuses on describing the method of analysis, which has few contributions, as the author recognizes, since it is based on a model already in use, while it does not provide a clear explanation of the andalusian university system in order to present the reality to be studied.  

The presentation of the results is unclear, with much emphasis on the variables and on demonstrating the validity of the analysis models and not so much on the presentation of the results obtained, which is slightly addressed in the conclusions.  What is interesting about this section are the future avenues of research.  

In addition, I have my doubts that the reference list complies with the journal's guidelines, so a thorough review is recommended.  

In my view, the presentation of results should be improved in order to be able to make a deeper analysis and draw more final conclusions. 

Author Response

Dear review, 

I resubmit the research paper entitled: “Measuring the Image of Private University as a generic product: Validation of a Scale” including all the changes and revisions you consider improving the manuscript. Thank you very much for your consideration.

We have been working really hard to present you this version. Please, find detailed the changes made in the attached document.

Thank you again,

The authors

Reviewer 2 Report

It is necessary to distribute elements from the methodology section to the results section

Make better use and explanation of the model figure

The 2 phases mentioned must be made explicit in the results

It is necessary to add a discussion section that manages to reflect the added value of the research, its practical implications and the contrast with authors

The introduction and literature review must incorporate references from the last 5 years (please update)

It is necessary that the conclusions do not include citations

Author Response

(The authors gave the same response as above.)

Round 2

Reviewer 1 Report

The modifications made are noteworthy and demonstrate an effort on the part of the authors to take advantage of previous work. The new title is more in line with what is actually presented in the manuscript. The incorporation of new contents improves the initial proposal and increases the possible interest it could have, although the object of study of the research is still very small.

Reviewer 2 Report

Everything suggested in the review has been complied with

The authors adequately explained all changes